# When combinations of humans and AI are useful: A systematic review and meta-analysis

Michelle Vaccaro ●[1,2], Abdullah Almaatouq ●[1] & Thomas Malone ●[1]✉

Inspired by the increasing use of artificial intelligence (AI) to augment humans, researchers have studied human–AI systems involving different tasks, systems and populations. Despite such a large body of work, we lack a broad conceptual understanding of when combinations of humans and AI are better than either alone. Here we addressed this question by conducting a preregistered systematic review and meta-analysis of 106 experimental studies reporting 370 effect sizes. We searched an interdisciplinary set of databases (the Association for Computing Machinery Digital Library, the Web of Science and the Association for Information Systems eLibrary) for studies published between 1 January 2020 and 30 June 2023. Each study was required to include an original human-participants experiment that evaluated the performance of humans alone, AI alone and human–AI combinations. First, we found that, on average, human–AI combinations performed significantly worse than the best of humans or AI alone (Hedges' $g = -0.23$; 95% confidence interval, −0.39 to −0.07). Second, we found performance losses in tasks that involved making decisions and significantly greater gains in tasks that involved creating content. Finally, when humans outperformed AI alone, we found performance gains in the combination, but when AI outperformed humans alone, we found losses. Limitations of the evidence assessed here include possible publication bias and variations in the study designs analysed. Overall, these findings highlight the heterogeneity of the effects of human–AI collaboration and point to promising avenues for improving human–AI systems.

People increasingly work with artificial intelligence (AI) tools in fields including medicine, finance and law, as well as in daily activities such as travelling, shopping and communicating. These human–AI systems have tremendous potential given the complementary nature of humans and AI—the general intelligence of humans allows us to reason about diverse problems, and the computational power of AI systems allows them to accomplish specific tasks that people find difficult.

A large body of work suggests that integrating human creativity, intuition and contextual understanding with AI's speed, scalability and analytical power can lead to innovative solutions and improved decision-making in areas such as health care[1], customer service[2] and scientific research[3]. However, a growing number of studies reveal that human–AI systems do not necessarily achieve better results than the best of humans or AI alone. Challenges such as communication barriers, trust issues, ethical concerns and the need for effective

[1]MIT Center for Collective Intelligence, Sloan School of Management, Massachusetts Institute of Technology, Cambridge, MA, USA. [2]Institute for Data, Systems, and Society, Schwarzman College of Computing, Massachusetts Institute of Technology, Cambridge, MA, USA. ✉e-mail: malone@mit.edu

**Fig. 1 | Forest plots of all effect sizes ($k$ = 370) included in the meta-analysis.** **a,b**, The positions of the points on the $x$ axes represent the values of the effect sizes, and the bars indicate the 95% CIs for the effect sizes. The colours of the points and bars correspond to the values of the effect sizes, with negative effect sizes coloured red and positive effect sizes coloured green. The black dashed line corresponds to an effect size of $g$ = 0, which means that the human–AI system performed the same as the baseline. The point at the bottom of the graph represents the meta-analytic average effect size and CI.

coordination between humans and AI systems can hinder the collaborative process[4–9].

These seemingly contradictory results raise important questions: when do humans and AI complement each other, and by how much? To address these issues, we conducted a systematic literature review and meta-analysis in which we quantified synergy in human–AI systems and identified factors that explain its presence (or absence) in different settings.

We focused on two outcomes: (1) human–AI synergy, where the human–AI group performs better than both the human alone and the AI alone, which is analogous to strong synergy in human groups[10,11]; and (2) human augmentation, where the human–AI group performs better than the human alone (see Supplementary Information section 1.1 for more details).

When evaluating human–AI systems, many studies focus on human augmentation[12–15]. This measure can serve important purposes in contexts for which full automation cannot happen for legal, ethical or safety reasons and in cases when AI does not align with human values. But when talking about the potential of human–AI systems, most people implicitly assume that the combined system should be better than either alone; otherwise, they would just use the best of the two[16]. They are thus looking for human–AI synergy. In light of these considerations, a growing body of work emphasizes evaluating and searching for synergy in human–AI systems[4,8,17–20].

To evaluate synergy in human–AI systems, we analysed 370 unique effect sizes from 106 different experiments published between January 2020 and June 2023 that included the performance of the human-only, AI-only and human–AI systems. On average, we found evidence of

human augmentation, meaning that the average human–AI systems performed better than the human alone. But we did not find human–AI synergy on average, meaning that the average human–AI systems performed worse than at least one of the human alone or the AI alone. So, in practice, if we consider only the performance dimensions the researchers studied, it would have been better to use either a human alone or an AI system alone rather than the human–AI systems studied.

While this overall result may appear discouraging, we also identified specific factors that did or did not contribute to synergy in human–AI systems. On the one hand, for example, much of the recent research in human–AI collaboration has focused on using AI systems to help humans make decisions by providing not only suggested decisions but also confidence levels or explanations. But we found that neither of these factors significantly affected the performance of human–AI systems.

On the other hand, little work has investigated the effects of task types and the relative performance of humans alone and AI alone. But we found that both factors significantly affected human–AI performance. Our work thus sheds needed light on promising directions for designing future human–AI systems to unlock the potential for greater synergy.

## Results

Our initial literature search yielded 5,126 papers, and, per the review process described in the Methods (see 'Literature review'), we identified 74 that met our inclusion criteria (Supplementary Fig. 1). These papers reported the results of 106 unique experiments, and many of the experiments had multiple conditions, so we collected a total of 370 unique effect sizes measuring the impact of human–AI collaboration

on task performance. Supplementary Fig. 2 highlights the descriptive statistics for the effect sizes in our analysis. We synthesized these data in a three-level meta-analytic model (see 'Data analysis' in Methods). We have made the materials required to reproduce our results publicly accessible through an Open Science Framework repository.

## Overall levels of human–AI synergy

In our primary analyses, we focused on human–AI synergy, so we compared the performance of the human–AI systems to a baseline of the human alone or the AI alone, whichever performed best. We found that the human–AI systems performed significantly worse overall than this baseline. The overall pooled effect was negative ($g = -0.23$; $t_{92} = -2.89$; two-tailed $P = 0.005$; 95% confidence interval (CI), −0.39 to −0.07) and considered small according to conventional interpretations[21].

However, when we compared the performance of the human–AI systems to a different baseline—the humans alone—we found substantial evidence of human augmentation. The human–AI systems performed significantly better than humans alone, and this pooled effect size was positive ($g = 0.64$; $t_{98} = 11.87$; two-tailed $P = 0.000$; 95% CI, 0.53 to 0.74) and medium to large[21]. Figure 1 displays a forest plot of these effect sizes. In other words, the human–AI systems we analysed were, on average, better than humans alone but not better than both humans alone and AI alone. For effect sizes that correspond to other potential outcomes of interest, see Supplementary Table 3 and Supplementary Fig. 3.

## Heterogeneity of human–AI synergy

We also found evidence for substantial heterogeneity of effect sizes in our estimations of human–AI synergy ($I^2 = 97.7\%$) and human augmentation ($I^2 = 93.8\%$) (see Supplementary Tables 5 and 6 for more details). Our moderator analysis identified characteristics of participants, tasks and experiments that led to different levels of human–AI synergy and human augmentation, and it helps explain sources of this heterogeneity. Figure 2 provides a visualization of the results of meta-regressions with our moderators. The definitions of the subgroups for different moderator variables are included in Supplementary Table 2, and more details of the regressions for other potential outcomes of interest are included in Supplementary Table 7.

First, we found that the type of task significantly moderated human–AI synergy ($F_{1,104} = 7.84$, two-tailed $P = 0.006$). Among decision tasks—those in which participants decided between a finite set of options—the pooled effect size for human–AI synergy was significantly negative ($g = -0.27$; $t_{104} = -3.20$; two-tailed $P = 0.002$; 95% CI, −0.44 to −0.10), which indicates performance losses from combining humans and AI. In contrast, among creation tasks—those in which participants created some sort of open-response content—the pooled effect size for human–AI synergy was positive ($g = 0.19$; $t_{104} = 1.35$; two-tailed $P = 0.180$; 95% CI, −0.09 to 0.48), pointing to synergy between humans and AI. Even though the average performance gains for creation tasks were not significantly different from 0 (presumably because of the relatively small sample size of $n = 34$), the difference between losses for decision tasks and gains for creation tasks was statistically significant. Relatedly, we found that the type of data involved in the task significantly moderated both human–AI synergy ($F_{4,101} = 15.24$, two-tailed $P = 0.000$) and human augmentation ($F_{4,101} = 6.52$, two-tailed $P = 0.000$).

Second, we found that the performance of the human and AI relative to each other impacted both human–AI synergy ($F_{1,104} = 81.79$, two-tailed $P = 0.000$) and human augmentation ($F_{1,104} = 24.35$, two-tailed $P = 0.000$). As shown in Fig. 2, when the human alone outperformed the AI alone, the combined human–AI system outperformed both alone with an average pooled effect size for human–AI synergy of $g = 0.46$ ($t_{104} = 5.06$; two-tailed $P = 0.000$; 95% CI, 0.28 to 0.66), a medium-sized effect[21]. But when the AI alone outperformed the human alone, performance losses occurred in the combined system relative

to the AI alone, with a negative effect size for human–AI synergy of $g = -0.54$ ($t_{104} = -6.20$; two-tailed $P = 0.000$; 95% CI, −0.71 to −0.37), a medium-sized effect[21]. Given the importance of this moderator, we fit separate meta-analytic models on the subsets of results where (1) the AI performed better alone and (2) the human performed better alone, and we report the results for human–AI synergy and human augmentation in Supplementary Table 4.

The performance of the human and AI relative to each other also affected the degree of human augmentation in the human–AI systems ($F_{1,104} = 24.35$, two-tailed $P = 0.000$). When the AI outperformed the human alone, greater performance gains tended to occur in the human–AI systems relative to the human alone, and the pooled effect size for human augmentation was positive and medium to large in magnitude ($g = 0.74$; $t_{104} = 13.50$; two-tailed $P = 0.000$; 95% CI, 0.63 to 0.85) (see Supplementary Figs. 4–7 for a more detailed visualization of this result for decision tasks).

We also found that the type of AI involved in the experiment ($F_{2,103} = 3.77$, two-tailed $P = 0.026$) and the year of publication ($F_{3,102} = 3.65$, two-tailed $P = 0.015$) moderated human–AI synergy, and the experimental design moderated human augmentation ($F_{1,104} = 4.90$, two-tailed $P = 0.029$). See Supplementary Fig. 8 for a more detailed visualization of the effect sizes by year of publication.

The remaining moderators we investigated were not statistically significant for human–AI synergy or human augmentation (explanation, confidence, participant type and division of labour).

## Discussion

Systems that combine human intelligence and AI tools can address multiple issues of societal importance, from how we diagnose disease to how we design complex systems[22–24]. But some studies show that augmenting humans with AI can lead to better outcomes than humans or AI working alone[24–26], while others show the opposite[4,7,9]. These seemingly disparate results raise two important questions: How effective is human–AI collaboration in general? And under what circumstances does this collaboration lead to performance gains versus losses? Our study analyses over three years of recent research to provide insights into both of these questions.

### Performance losses from human–AI collaboration

Regarding the first question, we found that, on average among recent experiments, human–AI systems did not exhibit synergy: the human–AI groups performed worse than either the human alone or the AI alone. This result complements the qualitative literature reviews on human–AI collaboration[27–29], which highlight some of the surprising challenges that arise when integrating human intelligence and AI. For example, people often rely too much on AI systems (overreliance), using its suggestions as strong guidelines without seeking and processing more information[6,30,31]. Other times, however, humans rely too little on AI (underreliance), ignoring its suggestions because of adverse attitudes towards automation[7,31,32].

Interestingly, we found that, among this same set of experiments, human augmentation did exist in the human–AI systems: the human–AI groups performed better than the humans working alone. Thus, even though the human–AI combinations did not achieve synergy on average, the AI system did on average help humans perform better. This result can occur, of course, because by definition the baseline for human–AI synergy is more stringent than that for human augmentation. It may also occur, however, because obtaining human–AI synergy requires different forms of human–AI interaction, or because the recent empirical studies were not appropriately designed to elicit human–AI synergy.

### Moderating effect of task type

With the large dataset we collected, we also performed analyses of factors that influence the effectiveness of human–AI collaboration. We

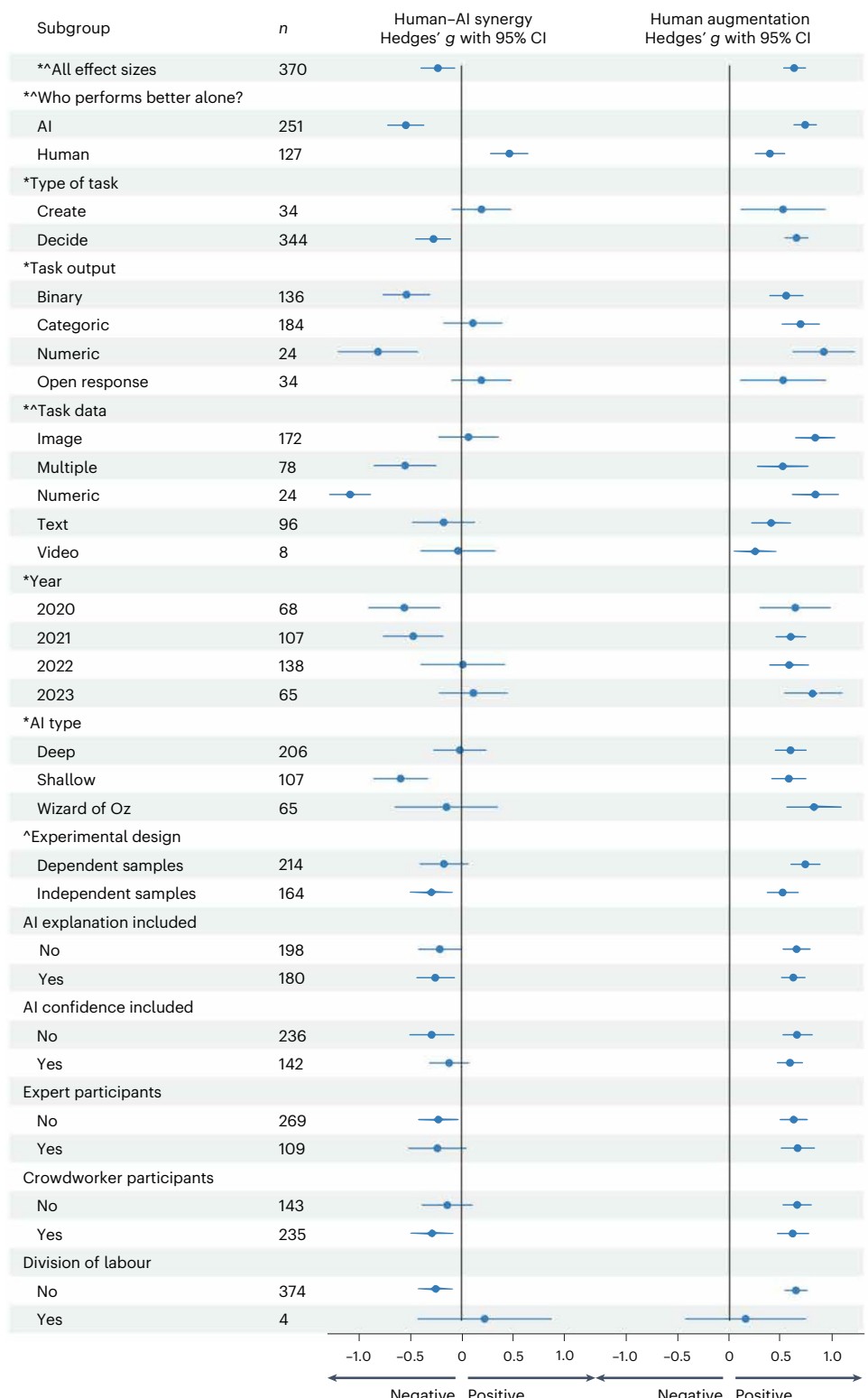

**Fig. 2 | Results from the three-level meta-regression models for the moderator variables.** Here *n* is the number of included effect sizes for the moderator subgroup level, and the estimated effect size with the corresponding 95% CI is shown for each level. The symbols in front of the moderators indicate whether there is a statistically significant difference between the subgroups for human–AI synergy (*) and human augmentation (^).

found that the type of task significantly moderated synergy in human–AI systems: decision tasks were associated with performance losses, and creation tasks were associated with performance gains.

We hypothesize that this advantage for creation tasks occurs because even when creation tasks require the use of creativity, knowledge or insight for which humans perform better, they often also involve substantial amounts of somewhat routine generation of additional content that AI can perform as well as or better than humans. For instance, generating a good artistic image usually requires some creative inspiration about what the image should look like, but it also often requires a fair amount of more routine fleshing out of the details of the image. Similarly, generating many kinds of text documents often

requires knowledge or insight that humans have and computers do not, but it also often requires filling in boilerplate or routine parts of the text as well.

With most of the decision tasks studied in our sample, however, both the human and the AI system make a complete decision, with the humans usually making the final choice. Our results suggest that with these decision tasks, better results might have been obtained if the experimenters had designed processes in which the AI systems did only the parts of the task for which they were clearly better than humans. Only 3 of the 100+ experiments in our analysis explore such processes with a predetermined delegation of separate subtasks to humans and AI. With the four effect sizes from these 3 experiments, we found that, on average, human–AI synergy ($g = 0.22$, $t_{104} = 0.69$; two-tailed $P = 0.494$; 95% CI, −0.42 to 0.87) occurred, but the result was not statistically significant (see Supplementary Information section 2.6 for a more detailed discussion of these experiments).

## Moderating effect of relative human/AI performance

Interestingly, when the AI alone outperformed the human alone, substantial performance losses occurred in the human–AI systems. When the human outperformed the AI alone, however, performance gains occurred in the human–AI systems. This finding shows that human–AI performance cannot be explained with a simple average of the human alone and AI alone. In such a case, human–AI synergy could never exist[33].

Most (>95%) of the human–AI systems in our dataset involved humans making the final decisions after receiving input from AI algorithms. In these cases, one potential explanation of our result is that, when the humans are better than the algorithms overall, they are also better at deciding in which cases to trust their own opinions and in which to rely more on the algorithm's opinions.

For example, Cabrera et al.[34] used an experimental design in which participants in the human–AI condition saw a problem instance, an AI prediction for that instance and, in some cases, additional descriptions of the accuracy of the AI in this type of instance. The same experimental design, with the same task interface, participant pool and accuracy of the AI system, was used for three separate tasks: fake hotel review detection, satellite image classification and bird image classification. For fake hotel review detection, the researchers found that the AI alone achieved an accuracy of 73%, the human alone achieved an accuracy of 55% and the human–AI system achieved an accuracy of 69%. In this case, we hypothesize that, since the people were less accurate, in general, than the AI algorithms, they were also not good at deciding when to trust the algorithms and when to trust their own judgement, so their participation resulted in lower overall performance than for the AI algorithm alone.

In contrast, Cabrera et al.[34] found that, for bird image classification, the AI alone achieved an accuracy of 73%, the human alone achieved an accuracy of 81% and the human–AI system achieved an accuracy of 90%. Here, the humans alone were more accurate than the AI algorithms alone, so we hypothesize that the humans were good at deciding when to trust their own judgements versus those of the algorithms, and the overall performance thus improved over either humans or AI alone.

## Surprisingly insignificant moderators

We also investigated other moderators such as the presence of an explanation, the inclusion of the confidence of the AI output and the type of participant evaluated. These factors have received much attention in recent years[4,24,26,35]. Given our result that, on average across our 300+ effect sizes, they do not impact the effectiveness of human–AI collaboration, we think researchers may wish to de-emphasize this line of inquiry and instead shift focus to the significant and less researched moderators we identified: the baseline performance of the human and AI alone, the type of task they perform, and the division of labour between them.

## Limitations

We want to highlight some general limitations of our meta-analytic approach to aid with the interpretation of our results. First, our quantitative results apply to the subset of studies we collected through our systematic literature review. To evaluate human–AI synergy, we required that papers report the performance of (1) the human alone, (2) the AI alone and (3) the human–AI system. We can, however, imagine tasks that a human and/or AI cannot perform alone but can when working with the other. Our analysis does not include such studies.

Second, we calculated effect sizes that correspond to different quantitative measures such as task accuracy, error and quality. By computing Hedges' $g$, a unitless standardized effect size, we can describe important relations among these experiments in ways that make them comparable across different study designs with different outcome variables[36]. The studies in our dataset, though, come from different samples of people—some look at doctors[37–39], others at crowdworkers[4,6,34] and still others at students[13,40,41]—and this variation can limit the comparability of the effect sizes to a degree[36]. The measurement error can also vary across experiments. For example, some studies estimate overall accuracy on the basis of the evaluation of as many as 500 distinct images[25], whereas others estimate it on the basis of the evaluation of as few as 15 distinct ones[42]. As is typical for meta-analyses[43], in our three-level model, we weighted effect sizes as a function of their variance across participants, so we did not account for this other source of variation in measurement.

Third, although we did not find evidence of publication biases, it remains possible that they exist, which would impact our literature base and, by extension, our meta-analytic results. However, we expect that if there were a publication bias operating here, it would be a bias to publish studies that showed significant gains from combining humans and AI. And since our overall results showed the opposite, it seems unlikely that they are a result of publication bias.

Fourth, our results only apply to the tasks, processes and participant pools that researchers have chosen to study, and these configurations may not be representative of the ways human–AI systems are configured in practical uses of AI outside the laboratory. In other words, even if there is not a publication bias in the studies we analysed, there might be a research topic selection bias at work.

Fifth, the quality of our analysis depends on the quality of the studies we synthesized. We tried to control for this issue by only including studies published in peer-reviewed publications, but the rigour of the studies may still vary in degree. For example, studies used different attention check mechanisms and performance incentive structures, which can both affect the quality of responses and thus introduce another source of noise into our data.

Finally, we found a high level of heterogeneity among the effect sizes in our analysis. The moderators we investigated account for some of this heterogeneity, but much remains unexplained. We hypothesize that interaction effects exist between the variables we coded (for example, explanation and type of AI), but we do not have enough studies to detect such effects. There are also certainly potential moderators that we did not analyse. For example, researchers mostly used their own experimental platforms and stimuli, which naturally introduce sources of variation between their studies. As the human–AI collaboration literature develops, we hope future work can identify more factors that influence human–AI synergy and assess the interactions among them.

## A roadmap for future work: finding human–AI synergy

Even though our main result suggests that—on average—combining humans and AI leads to performance losses, we do not think this means that combining humans and AI is a bad idea. On the contrary, we think it just means that future work needs to focus more specifically on finding effective processes that integrate humans and AI. Our other results suggest promising ways to proceed.

**Develop generative AI for creation tasks.** In our broad sample of recent experiments, the vast majority (about 85%) of the effect sizes were for decision-making tasks in which participants chose among a predefined set of options. But in these cases we found that the average effect size for human–AI synergy was significantly negative. In contrast, only about 10% of the effect sizes researchers studied were for creation tasks—those that involved open-ended responses. And in these cases we found that the average effect size for human–AI synergy was positive and significantly greater than that for decision tasks. This result suggests that studying human–AI synergy for creation tasks—many of which can be done with generative AI—could be an especially fruitful area for research.

Much of the recent work on generative AI with human participants, however, tends to focus on attitudes towards the tool[44,45], interviews or think-alouds with participants[46–48], or user experience instead of task performance[49–51]. Furthermore, the relatively little work that does evaluate human–AI collaboration according to quantitative performance metrics tends to report only the performance of the human alone and that of the human–AI combination (not the AI alone)[52]. This limitation makes evaluating human–AI synergy difficult, as the AI alone may be able to perform the task at a higher quality and speed than the participants involved in the experiment (typically crowdworkers). We thus need studies that further explore human–AI collaboration across diverse tasks while reporting the performance of the human alone, AI alone and human–AI system.

**Develop innovative processes.** Additionally, as discussed in ref. 33, human–AI synergy requires that humans be better at some parts of a task, AI be better at other parts of the task and the system as a whole be good at appropriately allocating subtasks to whichever partner is best for that subtask. Sometimes that is done by letting the more capable partner decide how to allocate subtasks, and sometimes it is done by assigning different subtasks a priori to the most capable partner (see Supplementary Information section 2.6 for specific examples from experiments in our dataset). In general, to effectively use AI in practice, it may be just as important to design innovative processes for how to combine humans and AI as it is to design innovative technologies[53].

**Develop more robust evaluation metrics for human–AI systems.** Many of the experiments in our analysis evaluate performance according to a single measure of overall accuracy, but this measure corresponds to different things depending on the situation, and it omits other important criteria for human–AI systems. For example, as one approaches the upper bound of performance, such as 100% accuracy, the improvements necessary to increase performance usually become more difficult for both humans and AI systems. In these cases, we may wish to consider a metric that applies a nonlinear scaling to the overall classification accuracy and thus takes such considerations into account[54] (Supplementary Information section 1.2).

More importantly, there are many practical situations where good performance depends on multiple criteria. For instance, in many high-stakes settings such as radiology diagnoses and bail predictions, relatively rare errors may have extremely high financial or other costs. In these cases, even if AI can, on average, perform a task more accurately and less expensively than humans, it may still be desirable to include humans in the process if the humans are able to reduce the number of rare but very undesirable errors. One potential approach for situations like these is to create composite performance measures that incorporate the expected costs of various kinds of errors. The human augmentation measure described is also appropriate for these high-stakes settings.

In general, we encourage researchers to develop, employ and report more robust metrics that consider factors such as task completion time, financial cost and the practical implications of different types of errors. These developments will help us better understand the significance of improvements in task performance as well as the effects of human–AI collaborations.

**Develop commensurability criteria.** As researchers continue to study human–AI collaboration, we also urge the field to develop a set of commensurability criteria, which can facilitate more systematic comparisons across studies and help us track progress in finding areas of human–AI synergy. These criteria could provide standardized guidelines for key study design elements such as:

(1) Task designs: establish a set of benchmark tasks that involve human–AI systems
(2) Quality constraints: specify acceptable quality thresholds or requirements that the human, AI and human–AI system must meet
(3) Incentive schemes: outline incentive structures (for example, payment schemes and bonuses) used to motivate human participants
(4) Process types: develop a taxonomy of interaction protocols, user interface designs and task workflows for effective human–AI collaboration
(5) Evaluation metrics: report the performance of the human, AI and human–AI system according to well-defined performance metrics

To further promote commensurability and research synthesis, we encourage the field to establish a standardized and open reporting repository, specifically for human–AI collaboration experiments. This centralized database would host the studies' raw data, code, system outputs, interaction logs and detailed documentation, adhering to the proposed reporting guidelines. It would thus facilitate the replication, extension and synthesis of research in the field. For example, by applying advanced machine learning techniques on such a dataset, we could develop predictive models to guide the design of human–AI systems optimized for specific constraints and contexts. Additionally, it would provide a means to track progress in finding greater areas of human–AI synergy.

In conclusion, our results demonstrate that human–AI systems often perform worse than humans alone or AI alone. But our analysis also suggests promising directions for the future development of more effective human–AI systems. We hope that this work will help guide progress in developing such systems and using them to solve some of our most important problems in business, science and society.

## Methods

We conducted this meta-analysis in accord with the guidelines from Kitchenham[55] on systematic reviews, and we followed the standards set forth by the Preferred Reporting Items for Systematic Reviews and Meta-Analyses[56].

### Literature review

**Eligibility criteria.** Per our preregistration (https://osf.io/wrq7c/?view_only=b9e1e86079c048b4bfb03bee6966e560), we applied the following criteria to select studies that fit our research questions. First, the paper needed to present an original experiment that evaluated some instance in which a human and an AI system worked together to perform a task. Second, it needed to report the performance of (1) the human alone, (2) the AI alone and (3) the human–AI system according to some quantitative measure(s). We therefore excluded studies that reported the performance of the human alone but not the AI alone, and likewise we excluded studies that reported the performance of the AI alone but not the human alone. Following this stipulation, we also excluded pure meta-analyses and literature reviews, theoretical work, qualitative analyses, commentaries, opinions and simulations. Third, we required the paper to include the experimental design, the number of participants in each condition and the standard deviation of the outcome in each condition, or enough information

to calculate it from other quantities. Finally, we required the paper to be written in English.

**Search strategy.** Given the interdisciplinary nature of human–AI interaction studies, we performed this search in multiple databases covering conferences and journals in the computer sciences, information sciences and social sciences, as well as other fields. Through consultation with a library specialist in these fields, we decided to target the Association for Computing Machinery Digital Library, the Association for Information Systems eLibrary and the Web of Science Core Collection for our review. To focus on current forms of AI, we limited the search to studies published between 1 January 2020 and 30 June 2023.

To develop the search string, we began by distilling the facets of studies that evaluated the performance of a human–AI system. We required the following: (1) a human component, (2) a computer component, (3) a collaboration component and (4) an experiment component. Given the multidisciplinary nature of our research question, papers published in different venues tended to refer to these components under various names[8,12,13,57,58]. For broad coverage, we compiled a list of synonyms and abbreviations for each component and then combined these terms with Boolean operations, resulting in the following search string:

(1) Human: human OR expert OR participant OR humans OR experts OR participants
(2) AI: AI OR 'artificial intelligence' OR ML OR 'machine learning' OR 'deep learning'
(3) Collaboration: collaborate OR assist OR aid OR interact OR help
(4) Experiment: 'experiment' OR 'experiments' OR 'user study' OR 'user studies' OR 'crowdsourced study' OR 'crowdsourced studies' OR 'laboratory study' OR 'laboratory studies'

The search term was (1) AND (2) AND (3) AND (4) in the abstract. See Supplementary Table 1 for the exact syntax for each database.

To further ensure comprehensive coverage, we also conducted a forward and backward search on all studies we found that met our inclusion criteria.

**Data collection and coding.** We conducted the search in each of these databases in July 2023. To calculate our primary outcome of interest—the effect of combining human intelligence and AI on task performance—we recorded the averages and standard deviations of the task performance of the human alone, the AI alone, and the human and AI working with each other, as well as the number of participants in each of these conditions (see Supplementary Information section 1.3 for details).

Many authors reported all of these values directly in the text of the paper. A notable number, however, reported them indirectly by providing 95% CIs or standard errors instead of the raw standard deviations. For these, we calculated the standard deviations using the appropriate formulas[59]. Additionally, multiple papers did not provide the exact numbers needed for such formulas, but the authors made the raw data of their study publicly accessible. In these cases, we downloaded the datasets and computed the averages and standard deviations using Python (v.3.11.9) or R (v.2023.06.0+421). If relevant data were only presented in the plots of a paper, we contacted the corresponding author to ask for the numeric values. If the authors did not respond, we used WebPlotDigitizer[60] to convert plotted values into numerical values. For papers that conducted an experiment that met our inclusion criteria but did not report all the values needed to calculate the effect size, we also contacted the corresponding author directly to ask for the necessary information. If the author did not respond, we could not compute the effect size for the study and did not include it in our analysis.

We considered and coded for multiple potential moderators of human–AI performance: (1) publication date, (2) preregistration status,

(3) experimental design, (4) data type, (5) task type, (6) task output, (7) AI type, (8) AI explanation, (9) AI confidence, (10) participant type and (11) performance metric. See Supplementary Table 2 for a description of each of these moderator variables. The additional information we recorded served descriptive and exploratory purposes.

Many papers conducted multiple experiments, contained multiple treatments or evaluated performance according to multiple measures. In such cases, we assigned a unique experiment identification number, treatment identification number and measure identification number to the effect sizes from the paper. Note that we defined experiments on the basis of samples of different sets of participants.

## Data analysis

**Calculation of effect size.** We computed Hedges' $g$ to measure the effect of combining human intelligence and AI on task performance[61]. For strong synergy, Hedges' $g$ represents the standardized mean difference between the performance of the human–AI system and that of the baseline, which can be the human alone or AI alone, whichever performs better on average. For human augmentation, Hedges' $g$ represents the standardized mean difference between the performance of the human–AI system and the baseline of the human alone.

We chose Hedges' $g$ as our measure of effect size because it is unitless, so it allows us to compare human–AI performance across different metrics, and it corrects for upward bias in the raw standardized mean difference (Cohen's $d$)[61]. See Supplementary Information section 1.4 for more details about this calculation.

**Meta-analytic model.** The experiments from the papers we analysed varied considerably. For example, they evaluated different tasks, recruited participants from different backgrounds and employed different experimental designs. Since we expected substantial between-experiment heterogeneity in the true effect sizes, for our analysis we used a random-effects model that accounts for variance in effect sizes that comes from both sampling error and 'true' variability across experiments[62].

Additionally, some of the experiments we considered generated multiple dependent effect sizes: they could involve multiple treatment groups, and they could assess performance according to more than one measure, for example. The more commonly used meta-analytic models assume independence of effect sizes, which makes them unsuitable for our analysis[63]. We thus adopted a three-level meta-analytic model in which effect sizes are nested within experiments, so we explicitly took the dependencies in our data into account in the model[63,64]. Furthermore, we used robust variance estimate methods to compute consistent estimates of the variance of our effect size estimates and, relatedly, standard errors, CIs and statistical tests, which account for the dependency of sampling errors from overlapping samples that occurred in experiments that compared multiple treatment groups to a single control group[65]. When evaluating the significance of our results, we applied the Knapp and Hartung adjustment and computed a test statistic, standard error, $P$ value and CI based on the $t$ distribution with $k - p$ degrees of freedom, where $k$ denotes the number of effect size clusters (that is, the number of experiments) and $p$ denotes the number of coefficients in the model. To perform our moderator analyses, we conducted separate meta-regressions for each of our moderator variables.

To interpret the degree of heterogeneity in our effect sizes, we calculated the popular $I^2$ statistic following ref. 66, which quantifies the percentage of variation in effect sizes that is not from sampling error. Furthermore, to distinguish the sources of this heterogeneity, we also calculated multilevel versions of the statistic, following ref. 64.

## Bias tests

In the context of our meta-analysis, publication bias may occur if researchers publish experiments that show evidence of significant

human–AI synergy more frequently than those that do not. Such actions would affect the data we collected and distort our findings. To evaluate this risk, we adopted multiple diagnostic procedures outlined by Viechtbauer and Cheung[67]. First, we created funnel plots that graph the observed effect sizes on the $x$ axis and the corresponding standard errors on the $y$ axis[68]. In the absence of publication bias, we expect the points to fall roughly symmetrically around the $y$ axis. We enhanced these plots with colours indicating the significance level of each effect size to help distinguish publication bias from other causes of asymmetry[69]. A lack of effect sizes in regions of statistical non-significance points to a greater risk of publication bias. We visually inspected the plots and performed Egger's regression test[70] as well as the rank correlation test[71] to evaluate the results in the funnel plots. These tests examine the correlation between the observed effect sizes and their associated sampling variances. A high correlation indicates asymmetry in the funnel plot, which may stem from publication bias.

Supplementary Fig. 9 displays the funnel plot of the included effect sizes, and we did not observe significant asymmetry or regions of missing data in the plot for our primary outcome, human–AI synergy. Egger's regression did not indicate evidence of publication bias in the sample ($\beta = -0.67$; $t_{104} = -0.78$; two-tailed $P = 0.438$; 95% CI, −2.39 to 1.04), nor did the rank correlation test ($\tau = 0.05$, two-tailed $P = 0.121$). Taken as a whole, these tests suggest that our results for human–AI synergy are robust to potential publication bias.

Importantly, however, we did find potential evidence of publication bias in favour of studies that report results in which the human–AI system outperforms the human alone (human augmentation). In this case, Egger's regression does point to publication bias in the sample ($\beta = 1.96$; $t_{104} = 3.24$; two-tailed $P = 0.002$; 95% CI, 0.76 to 3.16), as does the rank correlation test ($\tau = 0.19$, two-tailed $P = 0.000$). Note that we did not try to correct for potential publication bias to preserve the integrity of the original data and maintain transparency in our reporting. Many proposed adjustment methods can also lead to overcorrection and distort results[72].

The discrepancy between the symmetry in the funnel plot for human–AI synergy versus asymmetry in the funnel plot for human augmentation may reflect how many researchers and journals implicitly assume an interest in human augmentation, comparing the human–AI system to the human alone.

## Sensitivity analysis

For our primary analysis, we developed a three-level meta-analytic model that accounted for variance in the observed effect sizes (first level), variance between effect sizes from the same experiment (second level) and variance between experiments (third level). We then calculated cluster-robust standard errors, CIs and statistical tests for our effect size estimates in which we defined clusters at the level of the experiment. This model accounts for the dependency in effect sizes that result from evaluating more than one treatment against a common control group and assessing performance according to more than one measure. It does, however, consider the experiments in our analysis as independent from each other, even if they come from the same paper. We find this assumption plausible because the experiments recruited different sets of participants and entailed different tasks or interventions.

As a robustness check, though, we performed a sensitivity re-analysis in which we clustered at the paper level instead of the experiment level. This multilevel model accounts for variance in the observed effect sizes (first level), variance between effect sizes from the same paper (second level) and variance between papers (third level). We still calculated cluster-robust standard errors, CIs and statistical tests for our effect size estimates in which we defined clusters at the level of the experiment because the participant samples overlapped only on the level of the experiment. Using this approach, we found a comparable overall effect size for human–AI synergy ($g = -0.22$; $t_{67} = -2.46$;

two-tailed $P = 0.017$; 95% CI, −0.41 to −0.04) and for human augmentation ($g = 0.65$; $t_{69} = 9.96$; two-tailed $P = 0.000$; 95% CI, 0.52 to 0.78).

We also evaluated the robustness of our results to outlying and influential data points. To detect such data, we computed the residuals and Cook's distance for each effect size. We considered residual values greater or less than three standard deviations from the mean as outliers, and following ref. 73, we considered values greater than $4/n$ as high influence, where $n$ is the number of data points in our analysis. Using this approach, we identified 11 outliers for human–AI synergy and 9 outliers for human augmentation. We performed a sensitivity re-analysis on a dataset excluding these effect sizes, which resulted in similar effect sizes for human–AI synergy ($g = -0.25$; $t_{104} = -3.45$; two-tailed $P = 0.001$; 95% CI, −0.39 to −0.11) and human augmentation ($g = 0.60$; $t_{104} = 12.60$; two-tailed $P = 0.000$; 95% CI, 0.50 to 0.69).

Additionally, we conducted leave-one-out analyses, in which we performed a series of sensitivity re-analyses on the different subsets of effect sizes obtained by leaving out one effect size in our original dataset. We also conducted leave-one-out analyses at the experiment and publication levels. These tests show how each effect size, experiment and publication affect our overall estimate of the effect of human–AI collaboration on task performance. Summary effect sizes for human–AI synergy ranged from −0.28 to −0.19 with two-tailed $P < 0.05$ (0.000 to 0.019) in all cases, indicating the robustness of our results to any single effect size, experiment or paper; likewise, summary effect sizes for human augmentation ranged from 0.61 to 0.66 with two-tailed $P < 0.05$ (0.000 to 0.000) in all cases, indicating the robustness of our results to any single effect size, experiment or paper.

Lastly, we conducted a sensitivity re-analysis on a dataset that omits the effect sizes we estimated, using either WebPlotDigitizer or the information provided by the authors in their paper, and again we found almost identical results for human–AI synergy ($g = -0.21$; $t_{98} = -2.59$; two-tailed $P = 0.011$; 95% CI, −0.36 to −0.05) and human augmentation ($g = 0.64$; $t_{98} = 11.73$; two-tailed $P = 0.000$; 95% CI, 0.53 to 0.75).

We performed all quantitative analysis with the R statistical programming language, and we primarily relied on the package metafor[74].

### Reporting summary

Further information on research design is available in the Nature Portfolio Reporting Summary linked to this article.

## Data availability

We compiled the data used in this analysis based on the studies identified in our systematic literature review. We have made the data we collected available via the project's Open Science Framework repository (https://osf.io/wrq7c/?view_only=b9e1e86079c048b4bfb03bee6966e560). In our systematic review, we searched the following databases: the Association for Computing Machinery Digital Library (https://dl.acm.org/), the Web of Science (https://clarivate.com/webofscience-group/solutions/web-of-science/) and the Association for Information Systems eLibrary (https://aisnet.org/page/AISeLibrary).

## Code availability

We have shared the code used to conduct our analysis via the Open Science Framework repository (https://osf.io/wrq7c/?view_only=b9e1e86079c048b4bfb03bee6966e560).

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

## Acknowledgements

M.V. was funded by the Accenture Technology Convergence Fellowship. Additional funding to T.M. was provided by the Toyota Research Institute, by the MIT Quest for Intelligence and by the National Research Foundation, Prime Minister's Office, Singapore, under its Campus for Research Excellence and Technological Enterprise (CREATE) programme. The funders had no role in study design, data collection and analysis, decision to publish or preparation of the manuscript. We thank J. Kim and G. Campagna for their help collecting the data for this project, and we thank D. Eckles, M. Alsobay, R. Na and E. Hu for their helpful feedback.

## Author contributions

M.V. conceived the study idea with feedback from A.A. and T.M. M.V. collected the data and performed the statistical analysis, with feedback from A.A. and T.M. M.V., A.A. and T.M. wrote the manuscript.

## Competing interests

The authors declare no competing interests.

## Additional information

**Correspondence and requests for materials** should be addressed to Thomas Malone.

# Reporting Summary

## Statistics

For all statistical analyses, confirm that the following items are present in the figure legend, table legend, main text, or Methods section.

| n/a | Confirmed | |
|---|---|---|
| ☐ | ☒ | The exact sample size (*n*) for each experimental group/condition, given as a discrete number and unit of measurement |
| ☐ | ☒ | A statement on whether measurements were taken from distinct samples or whether the same sample was measured repeatedly |
| ☐ | ☒ | The statistical test(s) used AND whether they are one- or two-sided<br>*Only common tests should be described solely by name; describe more complex techniques in the Methods section.* |
| ☐ | ☒ | A description of all covariates tested |
| ☐ | ☒ | A description of any assumptions or corrections, such as tests of normality and adjustment for multiple comparisons |
| ☐ | ☒ | A full description of the statistical parameters including central tendency (e.g. means) or other basic estimates (e.g. regression coefficient) AND variation (e.g. standard deviation) or associated estimates of uncertainty (e.g. confidence intervals) |
| ☐ | ☒ | For null hypothesis testing, the test statistic (e.g. $F$, $t$, $r$) with confidence intervals, effect sizes, degrees of freedom and $P$ value noted<br>*Give P values as exact values whenever suitable.* |
| ☒ | ☐ | For Bayesian analysis, information on the choice of priors and Markov chain Monte Carlo settings |
| ☒ | ☐ | For hierarchical and complex designs, identification of the appropriate level for tests and full reporting of outcomes |
| ☐ | ☒ | Estimates of effect sizes (e.g. Cohen's *d*, Pearson's *r*), indicating how they were calculated |

*Our web collection on statistics for biologists contains articles on many of the points above.*

## Software and code

Policy information about availability of computer code

| | |
|---|---|
| Data collection | We identified relevant studies through our systematic literature review, which we describe in our Methods section. We extracted relevant effect sizes and experiment characteristics from these studies, and we used Excel (Version 16.87) for the data collection. |
| Data analysis | We performed all quantitative analysis with the R statistical programming language in RStudio (Version 2023.06.0+421). We used the following packages: metafor (Version 4.6-0), tidyverse (Version 2.0.0), broom (Version 1.0.6), gtsummary (Version 2.0.0), forestploter (Version 1.1.2), and pacman (Version 0.5.1). We share the code used in analyze the data in our project's OSF repository (https://osf.io/wrq7c/?view_only=b9e1e86079c048b4bfb03bee6966e560). |

For manuscripts utilizing custom algorithms or software that are central to the research but not yet described in published literature, software must be made available to editors and reviewers. We strongly encourage code deposition in a community repository (e.g. GitHub). See the Nature Portfolio guidelines for submitting code & software for further information.

## Data

Policy information about availability of data

All manuscripts must include a data availability statement. This statement should provide the following information, where applicable:

- Accession codes, unique identifiers, or web links for publicly available datasets
- A description of any restrictions on data availability
- For clinical datasets or third party data, please ensure that the statement adheres to our policy

We compiled the data used in this analysis based on the studies identified in our systematic literature review. We make the data we collected available via the project's Open Science Framework repository (https://osf.io/wrq7c/?view_only=b9e1e86079c048b4bfb03bee6966e560). In our systematic review, we searched the following databases: the ACM Digital Library (ACM DL) (https://dl.acm.org/), Web of Science (https://clarivate.com/webofsciencegroup/solutions/web-of-science/), and Association for Information Systems eLibrary (AISeL) (https://aisnet.org/page/AISeLibrary).

## Research involving human participants, their data, or biological material

Policy information about studies with human participants or human data. See also policy information about sex, gender (identity/presentation), and sexual orientation and race, ethnicity and racism.

| | |
|---|---|
| Reporting on sex and gender | Our study consists of a systematic review and meta-analysis of prior work. We did not collect information on sex and gender from the original studies. |
| Reporting on race, ethnicity, or other socially relevant groupings | Our study consists of a systematic review and meta-analysis of prior work. We did not collect information on race, ethnicity, or other socially relevant groupings from the original studies. |
| Population characteristics | Our study consists of a systematic review and meta-analysis of prior work. We collected the following population characteristics from the original studies: number of participants, whether the participant was a crowdworker, and whether the participant had domain expertise for the task. |
| Recruitment | Our study consists of a systematic review and meta-analysis of prior work, so we did not recruit any participants. |
| Ethics oversight | Our study consists of a systematic review and meta-analysis of prior work, so it did not require ethical approval. |

Note that full information on the approval of the study protocol must also be provided in the manuscript.

# Field-specific reporting

Please select the one below that is the best fit for your research. If you are not sure, read the appropriate sections before making your selection.

☐ Life sciences ☒ Behavioural & social sciences ☐ Ecological, evolutionary & environmental sciences

For a reference copy of the document with all sections, see nature.com/documents/nr-reporting-summary-flat.pdf

# Behavioural & social sciences study design

All studies must disclose on these points even when the disclosure is negative.

| | |
|---|---|
| Study description | Our study consists of a systematic review and meta-analysis of prior experiments that evaluate human-AI systems. For each of these prior experiments, we collect quantitative data about the performance of the human alone, AI alone, and human-AI system on the task. |
| Research sample | Our research sample consists of experiments identified in our systematic literature review. Given the interdisciplinary nature of human-AI interaction studies, we performed this search in multiple databases covering conferences and journals in the computer sciences, information sciences, and social sciences, as well as other fields. Through consultation with a library specialist in these fields, we decided to target the Association for Computing Machinery Digital Library (ACM DL), Association for Information Systems eLibrary (AISeL), and the Web of Science Core Collection (WoS) for our review. To further ensure comprehensive coverage, we also conducted a forward and backwards search on all studies we found that meet our inclusion criteria. Our initial literature search yielded 5126 papers, and, per the review process described in the Methods Section, we identified 74 that met our inclusion criteria. These papers reported the results of 106 unique experiments, and many of the experiments had multiple conditions, so we collected a total of 370 unique effect sizes measuring the impact of human-AI collaboration on task performance. Our sample is thus representative of the past experiments involving human-AI systems. If there was bias in the individual studies or the publication process, our sample may not be representative of the general public. We conduct a number of bias tests to evaluate this possibility (see Methods Section) and discuss the results and implications in our paper. |
| Sampling strategy | As discussed above, given the interdisciplinary nature of human-AI interaction studies, we performed this search in multiple databases covering conferences and journals in the computer sciences, information sciences, and social sciences, as well as other fields. Through consultation with a library specialist in these fields, we decided to target the Association for Computing Machinery |

Digital Library (ACM DL), Association for Information Systems eLibrary (AISeL), and the Web of Science Core Collection (WoS) for our review. To further ensure comprehensive coverage, we also conducted a forward and backwards search on all studies we found that meet our inclusion criteria.

**Data collection**

To calculate our primary outcome of interest -- the effect of combining human and artificial intelligence on task performance -- we recorded the averages and standard deviations of the task performance of the human alone, the AI alone, and the human and AI working with each other, as well as the number of subjects in each of these condition. We also considered and coded for multiple potential moderators of human-AI performance, namely: (1) publication date (2) pre-registration status (3) experimental design (4) data type (5) task type (6) task output (7) AI type (8) AI explanation (9) AI confidence (10) participant type and (11) performance metric. When collecting these data, the researchers were not blinded to the study hypotheses; however, we do not view this as a problem because the data extraction process was guided by a pre-registered protocol and relies entirely on published data.

**Timing**

To focus on current forms of artificial intelligence, we limited the search to studies published between January 1, 2020 and June 30, 2023. We conducted the search in July 2023.

**Data exclusions**

Per our pre-registration (see https://osf.io/wrq7c/?view_only=b9e1e86079c048b4bfb03bee6966e560), we applied the following criteria to select studies that fit our research questions. First, the paper needed to present an original experiment that evaluates some instance in which a human and an AI system work together to perform a task. Second, it needed to report the performance of (1) the human alone, (2) the AI alone, and (3) the human-AI system according to some quantitative measure(s). As such, we excluded studies that reported the performance of the human alone but not the AI alone, and likewise we excluded studies that reported the performance of the AI alone but not the human alone. Following this stipulation, we also excluded purely meta-analyses and literature reviews, theoretical work, qualitative analyses, commentaries, opinions, and simulations. Third, we required the paper to include the experimental design, the number of participants in each condition, and the standard deviation of the outcome in each condition, or enough information to calculate it from other quantities. Finally, we required the paper to be written in English. See our PRISMA Flow Diagram (Figure S1) for the number of studies excluded in the systematic review and meta-analysis and rationale for exclusion.

**Non-participation**

We did not recruit participants in our study.

**Randomization**

Our study consists of a systematic review and meta-analysis of prior work, so randomization was not applicable in our study. The data in our meta-analysis cover all of experiments identified in our systematic review.

# Reporting for specific materials, systems and methods

We require information from authors about some types of materials, experimental systems and methods used in many studies. Here, indicate whether each material, system or method listed is relevant to your study. If you are not sure if a list item applies to your research, read the appropriate section before selecting a response.

## Materials & experimental systems

| n/a | Involved in the study |
|-----|-----------------------|
| ☒ ☐ | Antibodies |
| ☒ ☐ | Eukaryotic cell lines |
| ☒ ☐ | Palaeontology and archaeology |
| ☒ ☐ | Animals and other organisms |
| ☒ ☐ | Clinical data |
| ☒ ☐ | Dual use research of concern |
| ☒ ☐ | Plants |

## Methods

| n/a | Involved in the study |
|-----|-----------------------|
| ☒ ☐ | ChIP-seq |
| ☒ ☐ | Flow cytometry |
| ☒ ☐ | MRI-based neuroimaging |

