## [Peer Review File · Nature Human Behaviour]

Peer Review Information

Journal: Nature Human Behaviour

Manuscript Title: When Are Combinations of Humans and AI Useful?- A Systematic Review and Meta-Analysis

Corresponding author name(s): Thomas Malone

Reviewer Comments & Decisions:

Decision Letter, initial version:

31st May 2023

Dear Dr. Malone,

Thank you once again for your manuscript, entitled "A Test for Evaluating Performance in Human-AI Systems", and for your patience during the peer review process.

Your Article has now been evaluated by 3 referees. You will see from their comments copied below that, although they find your work of potential interest, they have raised quite substantial concerns. In light of these comments, we cannot accept the manuscript for publication, but would be interested in considering a revised version if you are willing and able to fully address reviewer and editorial concerns.

We hope you will find the referees' comments useful as you decide how to proceed. If you wish to submit a substantially revised manuscript, please bear in mind that we will be reluctant to approach the referees again in the absence of major revisions. We are committed to providing a fair and constructive peer-review process. Do not hesitate to contact us if there are specific requests from the reviewers that you believe are technically impossible or unlikely to yield a meaningful outcome.

Editorially, we find reviewers' concerns about the generalizability of the findings to other tasks, the breadth of the meta-analysis, and the applicability to tail-cases in high-stakes settings particularly important. In a revision, we would therefore expect to see studies beyond 2021 included in your meta analysis, as well as a replications of Study 2/3 using a different task or tasks. Importantly, we ask that you address key Reviewer 3's concerns and think about how your test or a version of it could accommodate the cases where even rare errors in fast performing systems are important.

If you wish to submit a suitably revised manuscript, we would hope to receive it within 4 months. I would be grateful if you could contact us as soon as possible if you foresee difficulties with meeting this target resubmission date.

- Include a “Response to the editors and reviewers” document detailing, point-by-point, how you addressed each editor and referee comment. If no action was taken to address a point, you must provide a compelling argument. When formatting this document, please respond to each reviewer comment individually, including the full text of the reviewer comment verbatim followed by your response to the individual point. This response will be used by the editors to evaluate your revision and sent back to the reviewers along with the revised manuscript.
- Highlight all changes made to your manuscript or provide us with a version that tracks changes.

[REDACTED]

Thank you for the opportunity to review your work. Please do not hesitate to contact me if you have any questions or would like to discuss the required revisions further.

Sincerely,

[REDACTED]

Reviewer expertise:

Reviewer #1: computational science, AI in human behaviour

Reviewer #2: computer science

Reviewer #3: AI and its applications

REVIEWER COMMENTS:

Reviewer #1:

Remarks to the Author:

I very much enjoyed reading this paper. The manuscript introduces an innovative approach to quantifying the synergy between humans and AI systems, a topic of considerable importance in the current technological landscape. The authors' proposition of a "ratio of means" measure is a significant contribution to the field, offering a straightforward and elegant method for evaluating the performance of human-AI collaborations.

The authors' methodology, which includes a systematic review and analysis of papers published in 2021 and two studies involving the use of GPT-3 in creating HTML code, is well-structured and rigorous. Their exploration of a specific task (writing HTML code) with a state-of-the-art AI system (GPT-3) provides a concrete context for their research, which strengthens the clarity and applicability of their findings.

Moreover, the authors' discussion on the potential reasons for not finding higher ratios of improvement from combining people and computers is insightful. They suggest that achieving substantial synergies among people and computers may require new kinds of software, new ways of configuring groups, and more systematic attention from both computer science and social science researchers. This discussion offers valuable insights into the challenges and potential solutions for enhancing human-AI synergy, which could stimulate further research in this area.

However, there are several areas where the paper could be improved:

1. Scope of the Systematic Review: The authors' systematic review is limited to papers published in 2021. While this choice may reflect the rapid advancements in AI technology, it restricts the breadth of the analysis. Expanding the review to include papers from a wider range of years could provide a more comprehensive view of the field, potentially revealing important trends and developments over time.
2. Diversity of Tasks and AI Systems: The authors' studies focus on a specific task (writing HTML code) and a single AI system (GPT-3). While this focus allows for a detailed exploration of the task and the AI system, it may limit the generalizability of the findings. The authors could consider exploring a wider

range of tasks and AI systems to demonstrate the applicability and utility of their proposed measure.

3. Methodological Choices: The authors could provide more detailed explanations and justifications for their methodological choices. For instance, they could explain why they chose to focus on the task of writing HTML code and why they chose to use GPT-3. They could also discuss the potential limitations of these choices and how they might have influenced the results.

4. Acceptable Quality Constraints: In Study 2, the authors set an "acceptable quality constraint" of 80%, which seems somewhat arbitrary. It would be beneficial if the authors could provide a rationale for this choice. Furthermore, it's not clear what percentage of participants (or the AI) failed in the task in each condition. Providing this information could offer a more nuanced understanding of the performance of humans and AI systems in the task.

5. Alternative Approaches to Progress: The authors propose having contests similar to those sponsored by DARPA to stimulate progress in the development and use of human-computer systems. While this proposal is interesting, it might reflect a certain bias towards competitive approaches to progress, which might not be applicable or beneficial in all contexts. The authors could consider discussing other potential approaches to stimulating progress in the development and use of human-computer systems.

Despite these areas for improvement, the manuscript makes a valuable contribution to the field by introducing a promising approach to evaluate human-AI collaborations. The authors have laid a solid foundation for further research in this area. Their work could stimulate new lines of inquiry and potentially lead to advancements in the development and use of human-computer systems.

The authors are encouraged to build on this strong start and consider the suggestions for improvement to further strengthen their paper. By addressing these points, the authors could enhance the breadth, depth, and impact of their research, making a more substantial contribution to our understanding of human-AI synergy.

Reviewer #2:

Remarks to the Author:

I really like the first half of this article. This work contributes a form of meta-analysis conducted over human-AI interaction demonstrating that, actually, human-AI teams do not outperform AIs or teams alone. This null result is important for us to grapple with as a field.

The article should make much clearer that the framing research also finds and discusses this lack of improvement, and that what this paper concretely contributes is a quantitative measurement to

understand the nature of the (lack of) improvement. For example, prior work discussing the potential lack of complementarity:

Buçinca, Zana, et al. "Proxy tasks and subjective measures can be misleading in evaluating explainable ai systems." Proceedings of the 25th international conference on intelligent user interfaces. 2020.

Vasconcelos, Helena, et al. "Explanations Can Reduce Overreliance on AI Systems During Decision-Making." arXiv preprint arXiv:2212.06823 (2022).

Buçinca, Zana, Maja Barbara Malaya, and Krzysztof Z. Gajos. "To trust or to think: cognitive forcing functions can reduce overreliance on AI in AI-assisted decision-making." Proceedings of the ACM on Human-Computer Interaction 5.CSCW1 (2021): 1-21.

Lai, Vivian, Han Liu, and Chenhao Tan. ""Why is 'Chicago' deceptive?" Towards Building Model-Driven Tutorials for Humans." Proceedings of the 2020 CHI Conference on Human Factors in Computing Systems. 2020.

Zhang, Yunfeng, Q. Vera Liao, and Rachel KE Bellamy. "Effect of confidence and explanation on accuracy and trust calibration in AI-assisted decision making." Proceedings of the 2020 conference on fairness, accountability, and transparency. 2020.

Given that background, I think the paper can sharpen its argument by posing the question as an ongoing debate: does AI complement people? Can it? And by offering the meta-analysis, it can shed light on this question.

I would also strongly encourage the authors to adjust their framing to acknowledge the large and growing literature arguing for complementarity and synergy already. Gagan Bansal and colleagues have done much work in this area. Here are some examples:

Gagan Bansal, Tongshuang Wu, Joyce Zhou, Raymond Fok, Besmira Nushi, Ece Kamar, Marco Tulio Ribeiro, and Daniel Weld. 2021. Does the Whole Exceed its Parts? The Effect of AI Explanations on Complementary Team Performance. In Proceedings of the 2021 CHI Conference on Human Factors in Computing Systems (CHI '21). Association for Computing Machinery, New York, NY, USA, Article 81, 1–16. <https://doi.org/10.1145/3411764.3445717>

Bansal, Gagan, et al. "Beyond accuracy: The role of mental models in human-AI team performance." Proceedings of the AAAI conference on human computation and crowdsourcing. Vol. 7. 2019.

Bansal, Gagan, et al. "Updates in human-ai teams: Understanding and addressing the

performance/compatibility tradeoff." Proceedings of the AAAI Conference on Artificial Intelligence. Vol. 33. No. 01. 2019.

Bansal, Gagan, et al. "Is the most accurate ai the best teammate? optimizing ai for teamwork." Proceedings of the AAAI Conference on Artificial Intelligence. Vol. 35. No. 13. 2021.

Wilder, Bryan, Eric Horvitz, and Ece Kamar. "Learning to complement humans." arXiv preprint arXiv:2005.00582 (2020).

The authors do cite some subset of the work in both of these lists. What I'm mostly advocating for here are changes to the motivation and framing to more directly address the main claims made in this work.

I'd recommend downplaying the idea of a ratio as a deep methodological contribution, and focus on the results of the meta-analysis, which seem more durable. I'd love to see a correlational analysis of the features of the studies that do, and don't, seem to give rise to complementarity.

I found Study 2 less compelling, and would recommend cutting it and putting more detail into the meta-review. It doesn't connect deeply to the first part of the paper. As written, Study 2 feels like it is adding one more study of the same sort that the meta-review just analyzed. And, as a reader, it wasn't clear: why are we reading about this study now? What are we learning from this study, beyond what the meta-review just explored? Is it that GPT-3/4 will reconfigure the prior results?

To put it another way, Study 2 doesn't answer "why" or "when" --- there's not really a mechanism at play. It would make sense as a follow-on if the meta-analysis in Study 1 argued through a correlational analysis the factors associated with positive outcomes, and then Study 2 isolated them and tested them causally. Instead, it feels like the operational question of Study 2 is currently "Would it persist with GPT-3?", but with more space and time dedicated to it. It feels less novel.

Overall, I am hopeful that the authors can highlight the best parts of this work, and would love to see that published.

Reviewer #3:

Remarks to the Author:

Summary: This paper studies human-AI synergy by comparing human-AI performance vs. $\max(\text{human-only performance, AI-only performance})$. They perform 3 studies. First, they look at 25 recent studies that document human-AI collaboration and find that collaboration often doesn't carry significant (or often, any) benefit relative to the AI system alone. Next, they suggest that the powerful new generation

of technology (specifically, large language models) may have changed this scenario, and made human-AI collaboration more synergistic. To test this, they perform two studies where they pair human programmers and non-programmers with GPT-3, demonstrating that they can effectively code together despite being less efficient individually.

Comments: Human-AI collaboration is an important topic and I was glad to see work on it. However, I'm not sure I agree with the authors' perspective. As the authors observe, I think it is often the case that either the human or AI dominates in average performance at a given task, and therefore their collaboration is unlikely to yield any improvement on average performance. For example, recent work has shown that algorithms can dominate humans on bail predictions, radiology diagnosis, etc. The reason we do not implement these algorithms in isolation and insist on human-AI collaboration in high-stakes settings is because we are worried about tail performance (e.g., errors in edge cases, robustness). The proposed ratio does not capture this, and it's unclear if the recent studies the authors looked at focused on evaluating such robustness or just overall performance.

In fact, the argument made in the second and third studies also suffers this issue. As the authors note, GPT-4 is already able to code faster and better than many humans, so I suspect the authors' ratio for these studies may now be less than 1, matching prior studies. But we will likely continue to see human supervision of the outputs of these algorithms in the near future because even errors at a 1% or 0.1% rate can cripple software systems. Again, the authors' ratio doesn't capture this fundamental benefit of human-AI collaboration. In their study with GPT-3, their performance metric is speed given an "acceptable quality constraint (>80% correct submissions)." AI with human supervision will naturally be much slower than AI alone, and future iterations of GPT-4 will likely be correct much more than 80% of the time, but human supervision can protect against tail errors. As technology progresses (and given the selection bias that studies tend to focus on tasks where AI can perform well), the overall performance ratio will rarely exceed 1, but I don't think this signifies the end of human-AI collaboration.

Author Rebuttal to Initial comments

Reviewer #1:

I very much enjoyed reading this paper. The manuscript introduces an innovative approach to quantifying the synergy between humans and AI systems, a topic of considerable importance in the current technological landscape. The authors' proposition of a "ratio of means" measure is a significant contribution to the field, offering a straightforward and elegant method for evaluating the performance of human-AI collaborations.

The authors' methodology, which includes a systematic review and analysis of papers published

in 2021 and two studies involving the use of GPT-3 in creating HTML code, is well-structured and rigorous. Their exploration of a specific task (writing HTML code) with a state-of-the-art AI system (GPT-3) provides a concrete context for their research, which strengthens the clarity and applicability of their findings.

Moreover, the authors' discussion on the potential reasons for not finding higher ratios of improvement from combining people and computers is insightful. They suggest that achieving substantial synergies among people and computers may require new kinds of software, new ways of configuring groups, and more systematic attention from both computer science and social science researchers. This discussion offers valuable insights into the challenges and potential solutions for enhancing human-AI synergy, which could stimulate further research in this area.

We appreciate the positive review and thank you for highlighting the aspects of our work that are noteworthy. Note that based on this thoughtful review, as well as those from other reviewers, we no longer include Study 2 in our paper. Instead, we shift our focus to a synthesis of existing experiments involving human-AI collaboration.

However, there are several areas where the paper could be improved:

1. **Scope of the Systematic Review:** The authors' systematic review is limited to papers published in 2021. While this choice may reflect the rapid advancements in AI technology, it restricts the breadth of the analysis. Expanding the review to include papers from a wider range of years could provide a more comprehensive view of the field, potentially revealing important trends and developments over time.

We expanded the review to include papers published in 2020, 2021, 2022, and 2023, which provides a much more comprehensive view of the field and enables us to undertake a more rigorous synthesis of the experimental findings. The analysis in the previous version of this paper included only 25 papers, but this revised version now includes 74 papers, which reported the results of 106 unique experiments. And since many of the experiments had multiple conditions, we collected a total of 370 unique effect sizes measuring the impact of human-AI collaboration on task performance. We believe this expanded sample size provides a much more solid basis for making generalizations.

2. **Diversity of Tasks and AI Systems:** The authors' studies focus on a specific task (writing

HTML code) and a single AI system (GPT-3). While this focus allows for a detailed exploration of the task and the AI system, it may limit the generalizability of the findings. The authors could consider exploring a wider range of tasks and AI systems to demonstrate the applicability and utility of their proposed measure.

Thank you for raising this important note. As noted above, we now include over 100 experiments in our meta-analysis, which allows us to explore human-AI synergy across a wider range of tasks and AI systems. We believe that this change greatly enhances the generalizability of our findings. Additionally, we now also conduct moderator analyses, which allows us to identify types of tasks that lead to different levels of human-AI synergy (see Figure 2).

3. Methodological Choices: The authors could provide more detailed explanations and justifications for their methodological choices. For instance, they could explain why they chose to focus on the task of writing HTML code and why they chose to use GPT-3. They could also discuss the potential limitations of these choices and how they might have influenced the results.

Since we have omitted Studies 2 and 3, this comment is no longer applicable. However, we agree that methodological choices can greatly influence experimental results, particularly the choice of the type of task and AI system under evaluation. We provide a detailed discussion about these choices when planning future studies that study human-AI synergy (see pages 9 and 10).

4. Acceptable Quality Constraints: In Study 2, the authors set an "acceptable quality constraint" of 80%, which seems somewhat arbitrary. It would be beneficial if the authors could provide a rationale for this choice. Furthermore, it's not clear what percentage of participants (or the AI) failed in the task in each condition. Providing this information could offer a more nuanced understanding of the performance of humans and AI systems in the task.

Since we have omitted Studies 2 and 3, this comment is no longer applicable in its current form.

5. Alternative Approaches to Progress: The authors propose having contests similar to those sponsored by DARPA to stimulate progress in the development and use of human-computer systems. While this proposal is interesting, it might reflect a certain bias towards competitive

approaches to progress, which might not be applicable or beneficial in all contexts. The authors could consider discussing other potential approaches to stimulating progress in the development and use of human-computer systems.

We have omitted the discussion of contests in this version of the paper. But we do suggest other potential approaches to stimulating progress in the development and use of human- AI systems in the final section of our discussion, “A Roadmap for Future Work: Finding Human-AI Synergy.” As one example, we encourage the field to develop a set of commensurability criteria and a centralized open reporting repository for experiments on human-AI collaboration. This community resource can facilitate more systematic comparisons across studies and help us track and stimulate progress in finding areas of human-AI synergy.

Despite these areas for improvement, the manuscript makes a valuable contribution to the field by introducing a promising approach to evaluate human-AI collaborations. The authors have laid a solid foundation for further research in this area. Their work could stimulate new lines of inquiry and potentially lead to advancements in the development and use of human-computer systems.

The authors are encouraged to build on this strong start and consider the suggestions for improvement to further strengthen their paper. By addressing these points, the authors could enhance the breadth, depth, and impact of their research, making a more substantial contribution to our understanding of human-AI synergy.

By considering the suggestions offered, we believe we enhanced the breadth and depth of our research and hope it can make a more substantial contribution to the field.

Reviewer #2:

Remarks to the Author:

I really like the first half of this article. This work contributes a form of meta-analysis conducted over human-AI interaction demonstrating that, actually, human-AI teams do not outperform AIs or teams alone. This null result is important for us to grapple with as a field.

Thank you for this comment. We decided to expand the focus on this first half of the article in our revision. In particular, we analyzed the relevant experiments published

between January 2020 and June 2023 that included the performance of the human only, AI only, and human-AI systems. In our previous submission, we only collected data from experiments published in 2022, so we now cover a much larger time period. We identified 74 papers that met our inclusion criteria (Figure S1). These papers reported the results of 106 unique experiments, and many of the experiments had multiple conditions, so there we collected a total of 370 unique effect sizes measuring the impact of human-AI collaboration on task performance.

The article should make much clearer that the framing research also finds and discusses this lack of improvement, and that what this paper concretely contributes is a quantitative measurement to understand the nature of the (lack of) improvement. For example, prior work discussing the potential lack of complementarity:

Buçinca, Zana, et al. "Proxy tasks and subjective measures can be misleading in evaluating explainable ai systems." Proceedings of the 25th international conference on intelligent user interfaces. 2020.

Vasconcelos, Helena, et al. "Explanations Can Reduce Overreliance on AI Systems During Decision-Making." arXiv preprint arXiv:2212.06823 (2022).

Buçinca, Zana, Maja Barbara Malaya, and Krzysztof Z. Gajos. "To trust or to think: cognitive forcing functions can reduce overreliance on AI in AI-assisted decision-making." Proceedings of the ACM on Human-Computer Interaction 5.CSCW1 (2021): 1-21.

Lai, Vivian, Han Liu, and Chenhao Tan. "'Why is 'Chicago' deceptive?'" Towards Building Model-Driven Tutorials for Humans." Proceedings of the 2020 CHI Conference on Human Factors in Computing Systems. 2020.

Zhang, Yunfeng, Q. Vera Liao, and Rachel KE Bellamy. "Effect of confidence and explanation on accuracy and trust calibration in AI-assisted decision making." Proceedings of the 2020 conference on fairness, accountability, and transparency. 2020.

Given that background, I think the paper can sharpen its argument by posing the question as an ongoing debate: does AI complement people? Can it? And by offering the meta-analysis, it can shed light on this question.

We reframed our introduction to make clear that this line of research also finds and discusses this lack of improvement. For example, in paragraph two of the introduction, we draw upon this line of research when we say:

“Challenges such as communication barriers, trust issues, and the need for effective coordination between humans and AI systems can hinder the collaborative process [4, 5, 6, 7]. Additionally, the integration of AI into workflows may require difficult changes in organizational structures and processes [8] or raise ethical concerns regarding transparency, accountability, and bias [9].”

We compare these results to those that do find evidence of human-AI synergy, and we then pose a version of this suggested question:

“These seemingly contradictory results raise important questions: When do humans and AI complement each other? And by how much?”

Then, we performed our meta-analysis to shed light on both questions. We thank you for these suggestions, and we think these changes make for a stronger argument.

I would also strongly encourage the authors to adjust their framing to acknowledge the large and growing literature arguing for complementarity and synergy already. Gagan Bansal and colleagues have done much work in this area. Here are some examples:

Gagan Bansal, Tongshuang Wu, Joyce Zhou, Raymond Fok, Besmira Nushi, Ece Kamar, Marco Tulio Ribeiro, and Daniel Weld. 2021. Does the Whole Exceed its Parts? The Effect of AI Explanations on Complementary Team Performance. In Proceedings of the 2021 CHI Conference on Human Factors in Computing Systems (CHI '21). Association for Computing Machinery, New York, NY, USA, Article 81, 1–16. <https://doi.org/10.1145/3411764.3445717>

Bansal, Gagan, et al. "Beyond accuracy: The role of mental models in human-AI team performance." Proceedings of the AAAI conference on human computation and crowdsourcing. Vol. 7. 2019.

Bansal, Gagan, et al. "Updates in human-ai teams: Understanding and addressing the

performance/compatibility tradeoff." Proceedings of the AAAI Conference on Artificial Intelligence. Vol. 33. No. 01. 2019.

Bansal, Gagan, et al. "Is the most accurate ai the best teammate? optimizing ai for teamwork." Proceedings of the AAAI Conference on Artificial Intelligence. Vol. 35. No. 13. 2021.

Wilder, Bryan, Eric Horvitz, and Ece Kamar. "Learning to complement humans." arXiv preprint arXiv:2005.00582 (2020).

The authors do cite some subset of the work in both of these lists. What I'm mostly advocating for here are changes to the motivation and framing to more directly address the main claims made in this work.

We also adjusted our framing to more explicitly acknowledge this growing literature. Specifically in the last sentence of paragraph 5:

"A growing body of work emphasizes evaluating and searching for strong synergy in human-AI systems [4, 15, 8, 16, 17, 18]."

Here, we cited the recommended papers, among others. We agree that this framing better motivates and addresses the main claims of our paper.

I'd recommend downplaying the idea of a ratio as a deep methodological contribution, and focus on the results of the meta-analysis, which seem more durable.

We agree with this suggestion, and we removed the ratio of means test from the paper, and instead use the more common Hedges' g as our measure of effect size.

Instead, we shift the focus to human-AI synergy, and we define two different types, in line with the literature on groups of humans. Specifically, in paragraph 4:

"Extending the concepts of strong and weak synergy from human teams [10, 11], we focused on two types of synergy: (1) strong synergy, where the human-AI group performs better than both the human alone and the AI alone; and (2) weak synergy, where the human-AI group performs better than either the human or the AI alone but not both."

I'd love to see a correlational analysis of the features of the studies that do, and don't, seem to give rise to complementarity.

We also added a moderator analysis of the features of the studies that do, and do not, give rise to human-AI synergy in our paper (see Figure 2 and Section 2.2 of the Results).

I found Study 2 less compelling, and would recommend cutting it and putting more detail into the meta-review. It doesn't connect deeply to the first part of the paper. As written, Study 2 feels like it is adding one more study of the same sort that the meta-review just analyzed. And, as a reader, it wasn't clear: why are we reading about this study now? What are we learning from this study, beyond what the meta-review just explored? Is it that GPT-3/4 will reconfigure the prior results?

To put it another way, Study 2 doesn't answer "why" or "when" --- there's not really a mechanism at play. It would make sense as a follow-on if the meta-analysis in Study 1 argued through a correlational analysis the factors associated with positive outcomes, and then Study 2 isolated them and tested them causally. Instead, it feels like the operational question of Study 2 is currently "Would it persist with GPT-3?", but with more space and time dedicated to it. It feels less novel.

We agree with this suggestion and removed Study 2 from the manuscript, which allows us to focus on a more detailed and comprehensive synthesis of the existing experiments in the field.

Overall, I am hopeful that the authors can highlight the best parts of this work, and would love to see that published.

Thank you for this very helpful review. By considering the suggestions offered, we believe we enhanced the breadth and depth of our research and hope it can make a more substantial contribution to the field.

Reviewer #3:

Remarks to the Author:

Summary: This paper studies human-AI synergy by comparing human-AI performance vs. max(human-only performance, AI-only performance). They perform 3 studies. First, they look

at 25 recent studies that document human-AI collaboration and find that collaboration often doesn't carry significant (or often, any) benefit relative to the AI system alone. Next, they suggest that the powerful new generation of technology (specifically, large language models) may have changed this scenario, and made human-AI collaboration more synergistic. To test this, they perform two studies where they pair human programmers and non-programmers with GPT-3, demonstrating that they can effectively code together despite being less efficient individually.

Comments: Human-AI collaboration is an important topic and I was glad to see work on it.

We agree that this topic is of pressing importance, and we thank you for your helpful review. By taking your comments into account, we believe we enhanced the breadth and depth of our research and hope it can make a more substantial contribution to the field.

However, I'm not sure I agree with the authors' perspective. As the authors observe, I think it is often the case that either the human or AI dominates in average performance at a given task, and therefore their collaboration is unlikely to yield any improvement on average performance.

Even though it is often the case that the human or AI dominates in average performance, many experiments find that collaboration results in significant performance improvements in the task(s) evaluated, as shown in Figure 1a. Here are some of the papers that report evidence of strong human-AI synergy:

- **Bansal, Gagan, et al. "Does the whole exceed its parts? The effect of AI explanations on complementary team performance." *Proceedings of the 2021 CHI conference on human factors in computing systems*. 2021.**
- **Weisz, Justin D., et al. "Better together? An evaluation of AI-supported code translation." *27th International conference on intelligent user interfaces*. 2022.**
- **Fügener, Andreas, et al. "Will humans-in-the-loop become borgs? Merits and pitfalls of working with AI." *Management Information Systems Quarterly (MISQ)*- Vol 45 (2021).**
- **Zhang, Qiaoning, Matthew L. Lee, and Scott Carter. "You complete me: Human-AI teams and complementary expertise." *Proceedings of the 2022 CHI conference on human factors in computing systems*. 2022.**

- Tejada, Heliodoro, et al. "AI-assisted decision-making: A cognitive modeling approach to infer latent reliance strategies." *Computational Brain & Behavior* 5.4 (2022): 491-508.

In our discussion, we address cases when collaboration may be more likely to result in performance gains relative to both the human alone and AI alone. For example, we note that:

“Additionally, as discussed in Donahue et al. (2023), human-AI synergy requires that humans be better at some parts of a task, AI be better at other parts of the task, and the system as a whole be good at appropriately allocating subtasks to whichever partner is best for that subtask. Sometimes that is done by letting the more capable partner decide how to do allocation of subtasks, and sometimes it is done by assigning different subtasks a priori to the most capable partner (see Section S3.3 of the SI for specific examples from experiments in our dataset). In general, to effectively use AI in practice, it may be just as important to design innovative processes for how to combine humans and AI as it is to design innovative technologies.” (page 9).

For example, recent work has shown that algorithms can dominate humans on bail predictions, radiology diagnosis, etc. The reason we do not implement these algorithms in isolation and insist on human-AI collaboration in high-stakes settings is because we are worried about tail performance (e.g., errors in edge cases, robustness). The proposed ratio does not capture this, and it’s unclear if the recent studies the authors looked at focused on evaluating such robustness or just overall performance.

In fact, the argument made in the second and third studies also suffers this issue. As the authors note, GPT-4 is already able to code faster and better than many humans, so I suspect the authors’ ratio for these studies may now be less than 1, matching prior studies.

But we will likely continue to see human supervision of the outputs of these algorithms in the near future because even errors at a 1% or 0.1% rate can cripple software systems. Again, the authors’ ratio doesn’t capture this fundamental benefit of human-AI collaboration. In their study with GPT-3, their performance metric is speed given an “acceptable quality constraint (>80% correct submissions).” AI with human supervision will naturally be much slower than AI alone,

and future iterations of GPT-4 will likely be correct much more than 80% of the time, but human supervision can protect against tail errors.

Since we have now focused the whole paper on a meta-analysis of over 74 other studies, we have removed the ratio of means test and the studies with GPT-3, so these comments are no longer directly applicable to the paper in its current form.

However, we believe that the point about edge cases in high stakes situations is important for many of the studies we analyzed, and we have now dealt with it in two ways:

- (1) On page 2, we defined measures of two different kinds of synergy: (a) *strong synergy*, where the human-AI group performs better than *both* the human alone and the AI alone; and (b) *weak synergy*, where the human-AI group performs better than *either* the human or the AI alone but not both. We also say that:

“When evaluating human-AI systems, many studies focus on a specific kind of weak synergy where the baseline is the performance of the humans alone [12, 13, 14, 15]. This measure can serve important purposes in contexts for which full automation cannot happen for legal, ethical, safety, or other reasons.”

We evaluate this type of synergy and summarize the results in Figure 1b and Figure 2.

- (2) On pages 9 and 10, we included a new section in the Discussion section called “Developing more robust evaluation criteria for human-AI systems.” This section includes the following paragraph that deals specifically with these issues:

“More importantly, there are many practical situations where good performance depends on multiple criteria. For instance, in many high-stakes settings such as radiology diagnoses and bail predictions, relatively rare errors may have extremely high financial or other costs. In these cases, even if AI can, on average, perform a task more accurately and less expensively than humans, it may still be desirable to include humans in the process if the humans are able to reduce the number of rare but very undesirable errors. One potential approach for situations like these is to create composite performance measures that incorporate the

expected costs of various kinds of errors. The weak synergy measure described is also appropriate for these high-stakes settings.”

As technology progresses (and given the selection bias that studies tend to focus on tasks where AI can perform well), the overall performance ratio will rarely exceed 1, but I don't think this signifies the end of human-AI collaboration.

We appreciate this thoughtful comment, and we agree that as AI capabilities continue advancing, we may see AI systems outperforming humans on an increasing number of tasks. But we also agree that these advances do not signify the end of human-AI collaboration. In the last section of our discussion, “A Roadmap for Future Work: Finding Human-AI Synergy,” we make this sentiment clear:

“Even though our main result suggests that – on average – combining humans and AI leads to performance losses, we do not think this means that combining humans and AI is a bad idea. On the contrary, we think it just means that future work needs to focus more specifically on finding effective processes that integrate humans and AI.”
(page 9)

The remainder of that section discusses other potential approaches to stimulating progress in the development and the use of human-AI systems. These suggestions apply to both strong human-AI synergy (where the human-AI systems outperform both the human and AI alone) and weak human AI synergy (where the human-AI system outperforms just the human alone). Ultimately, we advocate for a more concentrated effort on developing effective human-AI systems.

References

Bansal, Gagan, et al. "Does the whole exceed its parts? The effect of AI explanations on complementary team performance." *Proceedings of the 2021 CHI conference on human factors in computing systems*. 2021.

Donahue, Kate, Alexandra Chouldechova, and Krishnaram Kenthapadi. "Human-algorithm collaboration: Achieving complementarity and avoiding unfairness." *Proceedings of the 2022 ACM Conference on Fairness, Accountability, and Transparency*.

2022.

Fügener, Andreas, et al. "Will humans-in-the-loop become borgs? Merits and pitfalls of working with AI." *Management Information Systems Quarterly (MISQ)*-Vol 45 (2021).

Tejeda, Heliodoro, et al. "AI-assisted decision-making: A cognitive modeling approach to infer latent reliance strategies." *Computational Brain & Behavior* 5.4 (2022): 491-508.

Weisz, Justin D., et al. "Better together? An evaluation of AI-supported code translation." *27th International conference on intelligent user interfaces*. 2022.

Zhang, Qiaoning, Matthew L. Lee, and Scott Carter. "You complete me: Human-AI teams and complementary expertise." *Proceedings of the 2022 CHI conference on human factors in computing systems*. 2022.

Decision Letter, first revision:

13th August 2024

Dear Dr. Malone,

Thank you for your patience as we've prepared the guidelines for final submission of your Nature Human Behaviour manuscript, "When Are Combinations of Humans and AI Useful?" (NATHUMBEHAV-23031018A). Please carefully follow the step-by-step instructions provided in the attached file, and add a response in each row of the table to indicate the changes that you have made. Please also address the additional marked-up edits we have proposed within the reporting summary. Ensuring that each point is addressed will help to ensure that your revised manuscript can be swiftly handed over to our production team.

We would hope to receive your revised paper, with all of the requested files and forms within two-three weeks. Please get in contact with us if you anticipate delays.

If you have not done so already, please alert us to any related manuscripts from your group that are

under consideration or in press at other journals, or are being written up for submission to other journals (see: <https://www.nature.com/nature-research/editorial-policies/plagiarism#policy-on-duplicate-publication> for details).

Nature Human Behaviour offers a Transparent Peer Review option for new original research manuscripts submitted after December 1st, 2019. As part of this initiative, we encourage our authors to support increased transparency into the peer review process by agreeing to have the reviewer comments, author rebuttal letters, and editorial decision letters published as a Supplementary item. When you submit your final files please clearly state in your cover letter whether or not you would like to participate in this initiative. Please note that failure to state your preference will result in delays in accepting your manuscript for publication.

In recognition of the time and expertise our reviewers provide to Nature Human Behaviour's editorial process, we would like to formally acknowledge their contribution to the external peer review of your manuscript entitled "When Are Combinations of Humans and AI Useful?". For those reviewers who give their assent, we will be publishing their names alongside the published article.

Cover suggestions

We welcome submissions of artwork for consideration for our cover. For more information, please see our guide for cover artwork.

ORCID

Non-corresponding authors do not have to link their ORCIDs but are encouraged to do so. Please note that it will not be possible to add/modify ORCIDs at proof. Thus, please let your co-authors know that if they wish to have their ORCID added to the paper they must follow the procedure described in the following link prior to acceptance: <https://www.springernature.com/gp/researchers/orcid/orcid-for-nature-research>

Nature Human Behaviour has now transitioned to a unified Rights Collection system which will allow our Author Services team to quickly and easily collect the rights and permissions required to publish your

work. Approximately 10 days after your paper is formally accepted, you will receive an email in providing you with a link to complete the grant of rights. If your paper is eligible for Open Access, our Author Services team will also be in touch regarding any additional information that may be required to arrange payment for your article.

Please note that *Nature Human Behaviour* is a Transformative Journal (TJ). Authors may publish their research with us through the traditional subscription access route or make their paper immediately open access through payment of an article-processing charge (APC). Authors will not be required to make a final decision about access to their article until it has been accepted. Find out more about Transformative Journals

[REDACTED]

Best regards,

[REDACTED]

On behalf of

[REDACTED]

Reviewer #2:

Remarks to the Author:

I reviewed the first version of this article, and recommended that the authors focus on the metareview. I'm so glad that they did — this revision of the manuscript is strong, analytically clear, and conceptually useful to the human-AI interaction literature. I strongly recommend acceptance. My recommendation is focused on the importance of this meta-analysis to the large number of active research agendas that are seeking out complementarity in human-AI teams. The literature keeps banging its head against a wall--- and this meta-analysis begins to tell the story of why, and what we ought to do about it.

If most of the weak complementarity is due to a stronger AI pulling up the performance of a weaker human decision-maker, I'm a bit torn about how to communicate this result. Technically this is true of using Google too. And Word. And Photoshop. And PowerPoint. And nearly any other computational tool since Licklider first articulated his article of Man-Machine Symbiosis and Engelbart first advocated for augmenting human intellect with the NLS. I'm not sure that I would call it synergy, per se, even weak synergy. Synergy, definitionally as far as I know, requires both organisms/actors to benefit. Weak synergy, as defined in the paper, typically doesn't provide any benefit to the AI. So, I would encourage the authors to consider renaming "strong synergy" to "synergy" and "weak synergy" to something else that doesn't imply a mutual benefit, e.g., "augmentation". I leave the choice up to the authors.

Reviewer #4:

Remarks to the Author:

SUMMARY

This is a challenging review task that has resulted in a manuscript that has merit and makes a significant contribution. Being a new reviewer, I appreciate the care and attention the authors provided to their review response, which encompasses a substantial addition to and re-working of their manuscript. I present my comments below in relation to changes that I think classify as a suggestion, important, or critical to change. I want to underscore a couple of things. 1) it is not normally my inclination to describe manuscript features that are well done. Having done so several times below speaks highly to the methodological quality of this work. 2) While I would like to see all four cells of strong vs weak and human better vs AI better as a baseline in the supplement, I am ok if the authors don't land there. I think this is ready for publication with minor revisions.

INTRODUCTION

Consider adding to an already excellent rationale and context for this work (especially regarding tail performance and edge cases) that while we may strive to improve the performance of AI-human collaboration there are cases beyond performance where we may always require collaboration. We may also want to broaden our perspective of this measurement space and what is meant by performance. AI is sociopathic, there are applications where improved performance may not intersect with human values and ethics. For example, we saw demonstrable climate improvement during COVID-19 with

shutdown protocols. Human confinement and potentially more extreme AI informed solutions are not always desirable. An argument along these lines would also elevate the importance of the so-called weak synergy findings (suggest).

RESULTS/DISCUSSION

Strong vs weak and human alone and AI alone are all touched on which suggests we should care about a four-cell matrix: Strong vs both (human alone better), strong vs both (AI alone better), weak vs either (human alone), weak vs either (AI alone). We currently get Strong (grouped together, which masks important information), and weak (human alone) as our main analyses. Understandable to feature a subset of the results if it is too much for the main paper, but we should get the entire story in the supplemental materials. Currently we get only three of the four cells (important).

Even if this is not expanded in the main article, alerting readers to the 3 cells already in the supplement should be done (critical).

I do think in the supplement it would be good to break up Table 3S (perhaps along strong and weak) which should allow room to report I2 and Q for each subgroup analysis (critical).

Figure 1(a) it seems like the label should be reflexive and consistently “The human-AI group [under/out]performs the better of the human or AI alone” (suggest).

I do miss the trees in the forest because of the choice to present results and conclusion first. Some very important nuance and care by the authors is not transparent until we get to the methods. If this is the norm for Nature-Human Behavior then please change nothing. I would lean towards a more traditional organization so readers can see the important foundational work leading up to these findings (or skip them as desired). (suggest).

METHODS

As someone who has engaged in meta-analytic work since before some of these tools were available, I applaud the use of automation tools (such as the web plot digitizer). Also noteworthy is seeing the receipts for whether or not this substantively altered major analyses when left out (which is not the case).

For heterogeneity I do appreciate that we already have reporting for Tau2, I2, and Q. I would move this up in the supplemental materials so there is context for my suggestion regarding Table S3 (suggest).

Also of importance, the authors have appropriately and conscientiously accounted for dependencies in their analysis with a three level meta-analytic model and through an RVE analysis. They report heterogeneity of I2 at all three levels. This is fantastic, as is including why this is important in the main

text. Given the other role for an article in this context and outlet is to serve as an ambassador for meta-analysis I feel like the community is well represented by this work.

I applaud the use of funnel plots, egger's test, and rank correlation in determining publication bias and I also applaud the decision to not try and adjust for it when present in subsequent analyses. I do think that conscious choice should be disclosed with a rationale (suggest).

Table S3, the p value is a significance test for the hedges' g being different (better or worse) than the baseline, and I would subtly shift the language to reflect that. Especially germane if there is an additional p value reported for Q (but to avoid confusion I would probably just use a footnote for non-significant Q values). (suggest).

I am fascinated and would love to see brief discussion of the variation between years for strong synergy (suggest).

For a wider lay audience interested in human behavior that may not be versed in AI knowing that the core algorithms have not changed since 2017 but the LLM has grown over time could provide meaningful context to some of these results (including the variation by year) (suggest).

Author Rebuttal, first revision:

Reviewer #2 (Remarks to the Author):

I reviewed the first version of this article, and recommended that the authors focus on the metareview. I'm so glad that they did — this revision of the manuscript is strong, analytically clear, and conceptually useful to the human-AI interaction literature. I strongly recommend acceptance. My recommendation is focused on the importance of this meta-analysis to the large number of active research agendas that are seeking out complementarity in human-AI teams.

The literature keeps banging its head against a wall---and this meta-analysis begins to tell the story of why, and what we ought to do about it.

We thank you for your helpful recommendation in the first version of the article and appreciate your positive review. We agree that this revision can help the field understand the lack of complementarity in many human-AI systems and craft more effective research agendas for the future.

If most of the weak complementarity is due to a stronger AI pulling up the performance of a weaker human decision-maker, I'm a bit torn about how to communicate this result.

Technically this is true of using Google too. And Word. And Photoshop. And PowerPoint. And nearly any other computational tool since Licklider first articulated his article of Man-Machine Symbiosis and Engelbart first advocated for augmenting human intellect with the NLS. I'm not sure that I would call it synergy, per se, even weak synergy. Synergy, definitionally as far as I know, requires both organisms/actors to benefit. Weak synergy, as defined in the paper, typically doesn't provide any benefit to the AI. So, I would encourage the authors to consider renaming "strong synergy" to "synergy" and "weak synergy" to something else that doesn't imply a mutual benefit, e.g., "augmentation". I leave the choice up to the authors.

We thank you for this suggestion, and we decided to adopt your recommendation and rename "strong synergy" to "human-AI synergy" and "weak synergy" to "human augmentation" (see Introduction paragraph 4 and section S1.1 of the SI).

Reviewer #4 (Remarks to the Author):

SUMMARY

This is a challenging review task that has resulted in a manuscript that has merit and makes a significant contribution. Being a new reviewer, I appreciate the care and attention the authors provided to their review response, which encompasses a substantial addition to and re-working of their manuscript. I present my comments below in relation to changes that I think classify as a suggestion, important, or critical to change. I want to underscore a couple of things. 1) it is not normally my inclination to describe manuscript features that are well done. Having done so several times below speaks highly to the methodological quality of this work.

We appreciate the thoughtful and positive review and thank you for highlighting the aspects of our work that are noteworthy.

2) While I would like to see all four cells of strong vs weak and human better vs AI better as a baseline in the supplement, I am ok if the authors don't land there. I think this is ready for publication with minor revisions.

We agree that this information is useful, and we added the suggested table to the Supplementary Information (see Table S4 and Table S6). Note that following the recommendation of another reviewer, we decided to rename "strong synergy" to "human-AI synergy" and "weak synergy" to "human augmentation" (see Introduction paragraph 4 and section S1.1 of the SI). This slight reframing makes our definition of synergy consistent with the intuitive notion that synergy requires both actors – in our case humans and AI – to benefit from interaction.

INTRODUCTION

Consider adding to an already excellent rationale and context for this work (especially regarding tail performance and edge cases) that while we may strive to improve the performance of AI-human collaboration there are cases beyond performance where we may always require collaboration. We may also want to broaden our perspective of this measurement space and what is meant by performance. AI is sociopathic, there are applications where improved performance may not intersect with human values and ethics. For example, we saw demonstrable climate improvement during COVID-19 with shutdown protocols. Human confinement and potentially more extreme AI informed solutions are not always desirable. An argument along these lines would also elevate the importance of the so-called weak synergy findings (suggest).

Thank you for this suggestion – we agree that these are important cases in which we need human-AI collaboration, and we added a sentence highlighting them in our introduction:

“When evaluating human-AI systems, many studies focus on human augmentation [12-15]. This measure can serve important purposes in contexts for which full automation cannot happen for legal, ethical, or safety reasons and in cases when AI does not align with human values.” (Introduction, Paragraph 5)

RESULTS/DISCUSSION

Strong vs weak and human alone and AI alone are all touched on which suggests we should care about a four-cell matrix: Strong vs both (human alone better), strong vs both (AI alone better), weak vs either (human alone), weak vs either (AI alone). We currently get Strong (grouped together, which masks important information), and weak (human alone) as our main analyses. Understandable to feature a subset of the results if it is too much for the main paper, but we should get the entire story in the supplemental materials. Currently we get only three of the four cells (important).

We agree that this information is useful, and we added the suggested table to the Supplementary Information (see Table S4 and Table S6).

Even if this is not expanded in the main article, alerting readers to the 3 cells already in the supplement should be done (critical).

We also added a sentence alerting readers to this information in the main article.

“We fit separate meta-analytic models on the subset of results where (1) the

AI performs better alone and (2) the human performs better alone, and we report the results for human-AI synergy and human augmentation in Table S4.” (Section 2.2, Paragraph 3)

I do think in the supplement it would be good to break up Table 3S (perhaps along strong and weak) which should allow room to report I2 and Q for each subgroup analysis (critical).

We agree that this information is important, and we included it in Table S4 and Table S6.

Figure 1(a) it seems like the label should be reflexive and consistently “The human-AI group [under/out]performs the better of the human or AI alone” (suggest).

We appreciate this suggestion and adopted the recommended labeling.

I do miss the trees in the forest because of the choice to present results and conclusion first. Some very important nuance and care by the authors is not transparent until we get to the methods. If this is the norm for Nature-Human Behavior then please change nothing. I would lean towards a more traditional organization so readers can see the important foundational work leading up to these findings (or skip them as desired). (suggest).

Since this is the required formatting for *Nature Human Behavior*, we decided to leave the order as is.

METHODS

As someone who has engaged in meta-analytic work since before some of these tools were available, I applaud the use of automation tools (such as the web plot digitizer). Also noteworthy is seeing the receipts for whether or not this substantively altered major analyses when left out (which is not the case).

Thank you for your positive feedback on our use of automation tools like the WebPlotDigitizer in our meta-analysis. We also appreciate your recognition of the effort to validate the robustness of our analyses by checking for substantive changes when these tools were not utilized.

For heterogeneity I do appreciate that we already have reporting for Tau2, I2, and Q. I would move this up in the supplemental materials so there is context for my suggestion regarding Table S3 (suggest).

We appreciate this suggestion, and we moved the heterogeneity analysis toward the beginning of the additional results in the SI. Now, we report the heterogeneity analysis for full results (Table S5) before reporting the heterogeneity analysis for the subset of results in which the human performs better alone and the AI performs better alone (Table S6).

Also of importance, the authors have appropriately and conscientiously accounted for dependencies in their analysis with a three level meta-analytic model and through an RVE analysis. They report heterogeneity of I2 at all three levels. This is fantastic, as is including why this is important in the main text. Given the other role for an article in this context and outlet is to serve as an ambassador for meta-analysis I feel like the community is well represented by this work.

Thank you for this positive feedback, we also hope our work can serve as an ambassador for meta-analysis.

I applaud the use of funnel plots, egger's test, and rank correlation in determining publication bias and I also applaud the decision to not try and adjust for it when present in subsequent analyses. I do think that conscious choice should be disclosed with a rationale (suggest).

We thank you for this comment, and we added two sentences describing why we did not try to adjust for publication bias:

“Note that we did not try to correct for potential publication bias to preserve the integrity of the original data and maintain transparency in our reporting. Many proposed adjustment methods can also lead to overcorrection and distort results [73].” (Section 4.3, Paragraph 3).

Table S3, the p value is a significance test for the hedges' g being different (better or worse) than the baseline, and I would subtly shift the language to reflect that. Especially germane if there is an additional p value reported for Q (but to avoid confusion I would probably just use a footnote for non-significant Q values). (suggest).

We appreciate that you highlighted this potential ambiguity, and we added captions to Table S3, S4, S5, S6, and S7 describing the values in each of the tables.

I am fascinated and would love to see brief discussion of the variation between years for strong synergy (suggest).

We added a section about the variation in human-AI synergy and human augmentation over

time (see Section S2.7).

For a wider lay audience interested in human behavior that may not be versed in AI knowing that the core algorithms have not changed since 2017 but the LLM has grown over time could provide meaningful context to some of these results (including the variation by year) (suggest).

We appreciate you highlighting this point, and we included a discussion contrasting the evolution of LLMs and human-AI synergy in Section S2.7.

Final Decision Letter: